# Associations of COVID-19 Related Work Stressors with Psychological Distress: Racial and Ethnic Disparities in Californian Workers

**DOI:** 10.3390/ijerph20010144

**Published:** 2022-12-22

**Authors:** Timothy A. Matthews, Megan Guardiano, Negar Omidakhsh, Lara Cushing, Wendie Robbins, OiSaeng Hong, Jian Li

**Affiliations:** 1Department of Environmental Health Sciences, Fielding School of Public Health, University of California Los Angeles, Los Angeles, CA 90095, USA; 2School of Nursing, University of California Los Angeles, Los Angeles, CA 90095, USA; 3Independent Researcher, Los Angeles, CA 90095, USA; 4Occupational and Environmental Health Nursing Graduate Program, School of Nursing, University of California San Francisco, San Francisco, CA 94143, USA

**Keywords:** California, COVID-19, workers, work stressors, psychological distress, racial disparities

## Abstract

The COVID-19 pandemic continues to exert immense societal impacts, with recent data showing inequitable distribution of consequences among racial and ethnic groups. The objective of this study was to assess associations between COVID-related work stressors and psychological distress, with special emphasis on differences by race and ethnicity. Data were from the population-based California Health Interview Survey (CHIS) 2020. Associations of individual and cumulative work stressors, including job loss, reduced work hours, and working from home, with psychological distress in 12,113 workers were examined via multivariable linear regression, and stratified analyses were conducted for racial and ethnic subgroups. After adjustment for covariates, compared to workers with no work stressors, those who experienced either one or two/more work stressors had higher psychological distress (βs and 95% CIs were 0.80 [0.51, 1.09] and 1.98 [1.41, 2.56], respectively). Notably, experiencing cumulative (two/more) work stressors had much stronger effects on psychological distress among participants who were Black (β and 95% CI were 3.51 [1.09, 5.93]) or racial minorities (β and 95% CI were 3.57 [1.10, 6.05]). Occupational consequences of the COVID-19 pandemic were associated with increased psychological distress in Californian workers and inequitably distributed, with racial and ethnic minorities suffering the greatest burden.

## 1. Introduction

The COVID-19 pandemic continues to exert immense adverse impacts on physical and mental health worldwide, maintaining a formidable presence as an issue of major public health significance. Disruptions to quotidian life persist, subjecting broader society to restrictions such as social distancing, quarantine, and limited travel, as well as extreme economic fallout such as job losses and widescale shifts to working from home [1,2,3,4,5,6]. Such unprecedented changes to living and working conditions have resulted in dire consequences for the health and well-being of the population at large, including severe psychological distress [2,7,8]. In fact, the consequences of the COVID-19 pandemic extend far beyond the domains of living activities and even population health, exerting immense socio-economic challenges that inevitably result in amplifying the prevalence and influence of work-related stressors [9].

Certainly, an expansive literature has evidenced the deleterious impacts of occupational stressors across various dimensions. Empirical evidence demonstrates the interaction of adverse work-related events and psychosocial factors and mental health conditions ranging from psychological distress [5] and depression [1,10] to illicit drug use [11,12]. Such associations of work stressors with psychiatric pathology are subject to exacerbation amidst the prevailing social climate of the COVID-19 pandemic, with a plethora of studies documenting pronounced increases in the prevalence and severity of adverse health outcomes [3,5,7,13,14,15,16]. 

The working population as a whole has undoubtedly suffered pervasive and enduring consequences, especially with regard to employment conditions and worsened mental health. However, a rapidly developing literature has highlighted a concerning trend wherein the societal and health impacts of the COVID-19 pandemic appear to be inequitably distributed across population subgroups [5,8,16,17,18,19,20,21]. While occupational factors exert pronounced health impacts across all demographic strata, the effects within strata are not distributed equally. For instance, recent work has identified glaring health disparities across racial and ethnic groups, drawing attention to clear health and employment outcome gaps between Whites and people of color [5,8,16,19,22,23,24]. These data substantiate a predicament of increased vulnerability of racial and ethnic minority groups to the broad-scale impacts of the COVID-19 pandemic. Studies have shown that underprivileged populations—those most susceptible to adverse health effects due to their marginalized status—have been subjected to the heaviest occupational and disease burdens, including severely increased COVID-19 hospitalization and mortality [5,8,16,17,21,22,23,25]. Furthermore, the increase in hate crimes and racial discrimination during the pandemic has been associated with psychological distress and increased substance use behaviors [8]. Race is defined as a social construct indexing the multitude of intersectional “social, environmental, and structural factors for which race may serve as a proxy measure”, including experiences of racism [26]. The interaction of these factors with pre-existing disparities in COVID-19 exposure, susceptibility, and healthcare access has extensive implications for mental health [27], and hence, the aftermath of the COVID-19 pandemic is undeniably a pressing issue of social justice. There is an urgent need to examine and systematically investigate the imbalanced distribution of the COVID-19 pandemic’s effects on health and employment across subpopulations, with special attention to the role of race and ethnic identity.

Therefore, the objective of this study was to examine associations of work stressors, including job loss, reduced work hours, and working from home, with psychological distress in a large, population-based sample of U.S. workers who were working during the pandemic, using data from the California Health Interview Survey (CHIS) 2020. We hypothesize that work stressors are associated with psychological distress, and that cumulative work stressors will exhibit dose-dependent associations with psychological distress, such that workers with more cumulative work stressors will experience higher psychological distress. Furthermore, we hypothesize that associations of cumulative work stressors with psychological distress will be much stronger among racial and ethnic minority groups, indicating an increased susceptibility of marginalized populations to the occupational and mental health effects of the COVID-19 pandemic.

## 2. Materials and Methods

### 2.1. Study Population

Data were from the CHIS 2020 study [28]. The CHIS procedures were changed and adapted to meet the shifting demands and challenges presented by the COVID-19 pandemic in order to mount a rapid response. While the CHIS study design conventionally aggregates interviews conducted across weekly sample waves, the CHIS 2020 sampling strategy combined each month’s worth of weekly sample waves [29]. COVID-19 related questions were added in mid-March 2020, coinciding with the advent of shelter-in-place restrictions or stay-at-home orders in the state of California that resulted in the closure of schools and non-essential businesses [29,30]. Data were collected via online surveys and telephone interviews, and completion rates were higher in 2020 (11.4%) than in 2019 (8.6%) [30]. A total of 21,949 participants were surveyed, and we included 12,113 working participants in our study. All participants had full data on measures of COVID-19 related work stressors, sociodemographic variables, and psychological distress.

### 2.2. Measures

COVID-19 related work stressors encompassed job loss, reduced work hours, and working from home. Work stressors were assessed with the question “Have you experienced any of the following situations because of the Coronavirus or COVID-19 outbreak?”, with the following items: “I’ve lost my regular job” (job loss), “I’ve had a reduction in hours, or a reduction in income” (reduced work hours), and “I’ve switched to working from home” (working from home). These three work stressors were coded as a binary variable, “No” vs. “Yes”.

Psychological distress in the past 30 days was measured with the Kessler Psychological Distress Scale (K6), a widely used and validated self-report measure of moderate to severe psychological distress [31,32]. The K6 operationalizes psychological distress via six questions (example items: How often did you feel nervous? “How often did you feel worthless?”), with responses scored on a five-point Likert scale ranging from “none of the time” to “all of the time”. K6 values in the sample ranged from 0 to 24, with higher scores indicating higher psychological distress.

Data on sociodemographic factors and lifestyle behaviors were also assessed, including sex, age (18-34, 35-49; 50-64; and 65+), race (Hispanic or Latino; Non-Hispanic White; Non-Hispanic Asian; Non-Hispanic Black; and Non-Hispanic Other/American Indian/Alaskan Native/Two or more races), marital status (married; widowed/divorced/living with partner; never married), educational attainment (high school or less; some college; University degree or higher), annual household income (<$50,000; $50,000-99,999; ≥$100,000), insurance (insured; uninsured), and citizenship (U.S. born citizen; naturalized citizen; non-citizen).

### 2.3. Statistical Analysis

Weighting procedures designed for analysis of CHIS 2020 data were implemented to bring the characteristics of the sample in closer alignment with the sociodemographic attributes of the general Californian population. The weights applied to sample data compensated for probability of participant selection, addressed biases associated with participant non-response, adjusted for undercoverage in the sampling and survey process, and “reduced variance of the estimates by using auxiliary information” [33]. Replicate weights provided with the CHIS 2020 data were included in the regression models.

First, weighted descriptive statistics were generated, and relative frequencies were examined for characteristics of the sample population, in total and by race. Second, associations of individual job loss, reduced work hours, and working from home, as well as cumulative work stressors, with psychological distress were estimated independently via weighted multivariable linear regression, and the results were expressed as adjusted betas (βs) with 95% confidence intervals (CIs). Two-sided hypothesis testing was conducted at the significance level α=0.05. Multivariable models were calculated in two steps: Model I adjusted for age and sex; and Model II included further adjustment for marital status, educational attainment, household income, insurance, and citizenship status. In addition, we tested interactions between race and work stressors with psychological distress as the outcome to offer empirical foundations for race-specific analyses, and stratified analyses were conducted accordingly. We also tested for significant differences in exposure to work stressors by racial and ethnic group, with differences determined via weighted Chi-square and ANOVA tests. All analyses were performed using the SAS 9.4 software package, Survey Analysis Procedures.

## 3. Results

The weighted characteristics of the sample population are displayed in Table 1. The sample of 12,113 Californian participants was made up of roughly equal numbers of males and females, and most workers were in the younger to middle-age categories of 18-34 and 35-49 (35.55% and 32.37%, respectively), with some older workers 50-64 (25.59%) and 65+ (6.49%). The majority of participants were insured (92.35%) and had U.S. citizenship status (85.95%). Most participants had at least some college education, were married, and had an annual household income below $100,000. The racial and ethnic distribution of the sample was primarily Hispanic (41.19%) and included representation of Whites (35.46%), Asians (14.19%), Blacks (5.25%), and individuals with two or more racial identities or who were a racial or ethnic minority (Non-Hispanic Other/American Indian/Alaskan Native/Two or more races, 3.91%). Almost half the participants experienced one work stressor (49.29%), with fewer experiencing none (43.39%), and a limited number experienced two or more (7.32%). Working from home was the most prevalent work stressor (31.66%), followed by reduced work hours (22.89%) and job loss (9.88%). The distribution of the number of work stressors experienced was significantly different across racial and ethnic subgroups. In CHIS 2020, the mean level of psychological distress was 4.60, representing a substantial increase compared to previous years (see Figure 1) [34,35,36].

The results of the linear regression analyses for the aggregate sample are displayed in Table 2. The analyses showed significant associations of both individual and cumulative work stressors (defined as either one, or two/more work stressors) with psychological distress. Workers who experienced job loss had higher psychological distress (β and 95% CI = 1.18 [0.60, 1.76]), compared to those who did not lose their jobs. Additionally, workers who experienced reduced work hours had higher psychological distress (β and 95% CI = 0.87 [0.53, 1.22]), compared to those who did not have work hours reduced. Workers who experienced working from home also had higher psychological distress (β and 95% CI = 0.81 [0.53, 1.09]), compared to respondents who continued to work on-site. The analyses of cumulative stressors showed that compared to workers with no work stressors, those who experienced either one or two/more work stressors had higher psychological distress (βs and 95% CIs = 0.80 [0.51, 1.09] and 1.98 [1.41, 2.56], respectively).

The interaction analysis indicated a significant interaction between race and cumulative work stressors with psychological distress as the outcome (*p* < 0.05). Stratified analyses demonstrated differential responses to work stressors by racial and ethnic group, with results presented in Table 3 and Figure 2. Compared to Whites, Hispanic, and Asians, participants who were a racial or ethnic minority or two or more races who experienced two or more stressors had the highest psychological distress (β and 95% CI = 3.57 [1.10, 6.05]), followed by Black participants (β and 95% CI = 3.51 [1.09, 5.93]). White, Hispanic, and Asian participants who experienced two or more stressors showed increased psychological distress (βs and 95% CIs = 1.54 [0.99, 2.08], 2.20 [1.03, 3.78], and 2.00 [0.55, 3.45], respectively).

Regarding individual work stressors, participants who were a racial or ethnic minority or two or more races who experienced job loss also had the highest psychological distress (β and 95% CI = 2.69 [0.51, 4.88]). White and Hispanic participants who experienced job loss had increased psychological distress (βs and 95% CIs = 1.24 [0.49, 1.99] and 1.17 [0.16, 2.18], respectively). However, Asian participants showed a weak relationship between job loss and higher psychological distress. White, Hispanic, and Asian participants who experienced reduced work hours had increased psychological distress (βs and 95% CIs = 0.65 [0.24, 1.06], 1.08 [0.43, 1.73], and 1.04 [0.24, 1.83], respectively). Black participants showed null associations for reduced work hours. White and Asian participants who experienced working from home also showed increased psychological distress (βs and 95% CIs = 0.79 [0.47, 1.11] and 0.93 [0.31, 1.56], respectively), whereas Hispanic participants showed non-significant associations. Black participants who experienced working from home had the highest psychological distress (β and 95% CI = 2.62 [1.14, 4.09]), compared to all other participants who did not experience working from home.

## 4. Discussion

In this large, population-based, cross-sectional study of 12,113 Californian workers conducted near the onset of the COVID-19 pandemic, individual and cumulative work stressors were associated with increased psychological distress. Workers who experienced job loss, reduced work hours, and working from home exhibited significantly increased psychological distress. Furthermore, we conducted targeted analyses to elicit the potential impacts of race and ethnicity in associations of COVID-19 related work stressors with psychological distress. Stratified analyses by racial and ethnic category demonstrated differential impacts of work stressors with psychological distress, with racial and ethnic minority groups experiencing the greatest psychological distress compared to White, Hispanic, Black, and Asian racial groups. Notably, Black participants also exhibited higher psychological distress. Therefore, our findings provided support for our hypotheses. Our results showing increased psychological distress among workers working from home are comparable to those reported previously [3]; however, they contrast with findings from other cohorts demonstrating decreased psychological distress [37], or null effects [4]. Our overall findings are consistent with the general literature showing marked increases in psychological distress among U.S adults during the COVID-19 pandemic [7], as well as studies examining relationships of occupational stress with mental health symptoms [3,5,6]. This is the first study to examine associations of COVID-19-related work stressors with psychological distress among different racial and ethnic groups in California.

Our most compelling findings are for racial and ethnic groups. Associations of COVID-19 work stressors and psychological distress among participants who were a racial or ethnic minority were clearly—drastically—stronger compared to the White, Hispanic, or Asian racial groups. These results indicate conditions beyond identification gaps between White populations and people of color in mental health outcomes and highlight the radically increased vulnerability of minority populations to psychological distress in the context of cumulative work stressors. Furthermore, the observed significant differences in exposure to work stressors across racial and ethnic categories add to the weight of evidence outlining a constellation of racial disparities in health, wherein vulnerable populations experience not only exacerbated outcome prevalence and effect sizes, but also increased exposure prevalence [38]. Our findings are in concert with recent work examining racial disparities amidst the COVID-19 pandemic and add strength to the body of literature substantiating the disproportionate distribution of its societal impacts [5,8,16,19,20,21,22,23,24,25]. A systematic review of mental health outcomes using international data from over 300,000 participants found that being part of a racial or ethnic marginalized group predicted mental health inequalities during the COVID-19 pandemic [23]. These findings were replicated in the nationally representative National Health Interview Survey (NHIS) 2019 and the 2020-2021 Household Pulse Survey, where racial and ethnic minorities experienced significant increases in depression and anxiety compared to Whites [24]. Similarly, a study using data from the nationally representative Health, Ethnicity, and Pandemic (HEAP) Study found that associations of negative employment changes, such as job losses and pay cuts, with psychological distress were most pronounced in Black and Asian participants [5]. Another study using the same HEAP data demonstrated a higher prevalence of experienced racial discrimination during the COVID-19 pandemic among racial and ethnic minorities, and that these experiences of racial discrimination were associated with elevated psychological distress [8]. Participants of the HEAP study who experienced racial discrimination were also more likely to delay or forgo healthcare during the pandemic—such barriers to treatment and access to psychiatric care have been shown to be drivers of healthcare disparities [17,21]. In a study of over 1.5 million U.S. participants, White respondents were most likely to receive professional mental health care both before and during the pandemic, while in comparison, minority participants demonstrated lower levels of mental health service utilization [24].

The presence of neuropsychiatric symptoms prior to the COVID-19 pandemic has been shown to predict increased psychological distress during the pandemic [39]. Racial and ethnic minority populations were already experiencing increased psychosocial strain relative to White populations prior to the pandemic due to a confluence of sociopolitical and economic elements. Such factors include generally lower incomes and poverty, experiences of racial discrimination and prejudice, and limited social mobility due to occupational and educational constraints [8,16,18,19,20,21,22,23,25,27]. Data collected during the COVID-19 pandemic lockdown period found that experiences of racial discrimination were more prevalent among racial and ethnic minorities, and that these experiences of racism were also correlated with indicators of socioeconomic status such as low educational attainment and lack of internet access [18]. Minority groups were at risk for experiencing COVID-19 exposure and barriers to healthcare and COVID-19 testing accessibility, possibility in relation to a higher likelihood of residency in crowded, lower income areas [19]. Data from previous infectious disease outbreaks also show evidence of significant race and ethnicity-related disparities in potential risk, due to systematic differences in exposure, susceptibility, and access to healthcare [27]. Furthermore, these findings must also be interpreted in context of devastating rates of COVID-19 hospitalization and mortality among minority populations compared to Whites [16,20,21,22,25]. Hence, the initial state of vulnerability and high stress combined with the sudden onset of the multidimensional and far-reaching disturbances produced by the COVID-19 pandemic may in part explain the elevated psychological distress observed in Black and racial and ethnic minorities. The data indicate that the pandemic added further stress burden to already distressed populations, driving overall increases in psychiatric symptomatology and critically, widening already existing disparities between racial and ethnic subgroups.

We must also consider the potential impact of other work stressors beyond those explored in the present study; while we were limited to analyses of the major work stressors of job loss, reduced work hours, and working from home, there are additional work-related psychosocial exposures that may exert influences on workers’ mental health. For instance, theoretical frameworks such as Lazarus and Folkman’s Transactional Stress model and Karasek’s Job Demand-Control-Support model posit key roles of psychological appraisal and social support in the modulation of stress responses [40,41]. Furthermore, the COVID-19 pandemic precipitated several other severe stressors, including perceived risk of infection and emotional fatigue [42]. Recent evidence has also demonstrated a pervasive impact of COVID-19 related stressors specific to racial and ethnic contexts. For instance, Chinese migrant workers who experienced racial discrimination reported concealing infection symptoms and a fear of reporting illness, in conjunction with high job insecurity [43].

This study has several major strengths. Foremost is the timely nature of the data collection—the CHIS 2020 survey was administered during the acute phase of the COVID-19 pandemic, in the immediate period following the implementation of emergency stay-at-home orders mandating the closure of schools and non-essential businesses throughout California [30]. Almost all of the CHIS 2020 data (approximately 96%) were collected after the execution of these directives [29]. This allowed for the acute capture of the societal fallout related to the pandemic, i.e., widespread changes to working conditions and pronounced adverse impacts on mental health. Furthermore, the sample was large (*N* = 12,113) and represented residents in the state of California, with sufficient inclusion of multiple racial and ethnic groups; the CHIS is the largest health survey conducted at the state level and one of the largest nationally [30]. While the CHIS is specific to California, the large sample size and adequate representation of population subgroups increases the generalizability of the results. To leverage these key advantages of the data, the stratified statistical analyses were specifically designed to elicit potential health disparities between racial and ethnic groups and were ultimately able to demonstrate severe contrasts in mental health outcomes. Finally, most prior studies examining COVID-19 related employment stressors only assessed the effects of single exposures; our study is unique in its pioneering attempt at advancing methodology by including cumulative employment stressors. 

The prevailing limitation of this study is the cross-sectional study design, which prohibits analyses of causal inference and raises the possibility of reverse causation. Most critically, the length of unemployment associated with the impacts of COVID-19 were indeterminable; reported job losses may have been in the short term or in the long term. We are also unable to address potential underlying psychobiological mechanisms responsible for the observed increases in psychological distress among participants who experienced high cumulative work stressors. Our results may also be affected by response bias, as a report by the Social Science Research Solutions (SSRS) organization and UCLA Center for Health Policy Research identified differential response across key subgroups in CHIS 2020 [30]. Possible drivers of nonresponse included evictions and foreclosures, increased childcare burden, and disproportionate impacts on the elderly [30]. These data indicate that certain segments of the Californian population did not participate in CHIS 2020 due to experiencing greater COVID-19 related stress and sequelae. This suggests that in fact, the effect sizes presented in the present study understate the true strength of the associations. On the other hand, the lower proportions of Black participants in our sample (5.25%) and of individuals who experienced two or more work stressors (7.32%) may constitute an inherent limitation, as a greater degree of variance in sample composition could have augmented the statistical significance of the findings. Additionally, prior analyses of CHIS data reported “measurement nonequivalence in the K6 among racially/ethnically and linguistically diverse adults”, suggesting that our results regarding self-reported psychological distress may be subject to cultural differentiation of perception [44]. Notably, many large surveys in the U.S. have repeatedly identified a “Black-White mental health paradox” wherein Black participants exhibit lower levels of psychological distress [45,46,47]. Finally, due to a lack of data on lifestyle behaviors such as alcohol consumption and physical activity, we were unable to assess the potential role of behavioral factors in associations of work stressors with psychological distress. Physical activity is well-evidenced as a protective factor in physical health and mental conditions [48,49]. Prior studies examining the behavioral impacts of the COVID-19 pandemic in the U.S. reported that individuals generally engaged in less healthful and more unhealthy behaviors [39,50,51,52]. These lifestyle factors may contribute to psychological distress.

## 5. Conclusions

In this study of a population-based sample of U.S workers in California, cumulative work stressors—including job loss, reduced work hours, and working from home—were associated with increased psychological distress. Associations of work stressors with psychological distress were most pronounced in Black and racial and ethnic minority groups. These results underscore an urgent need for government and employer policy interventions that address a concerning trend of racial and ethnic health disparities in the wake of the COVID-19 pandemic.

## Figures and Tables

**Figure 1 ijerph-20-00144-f001:**
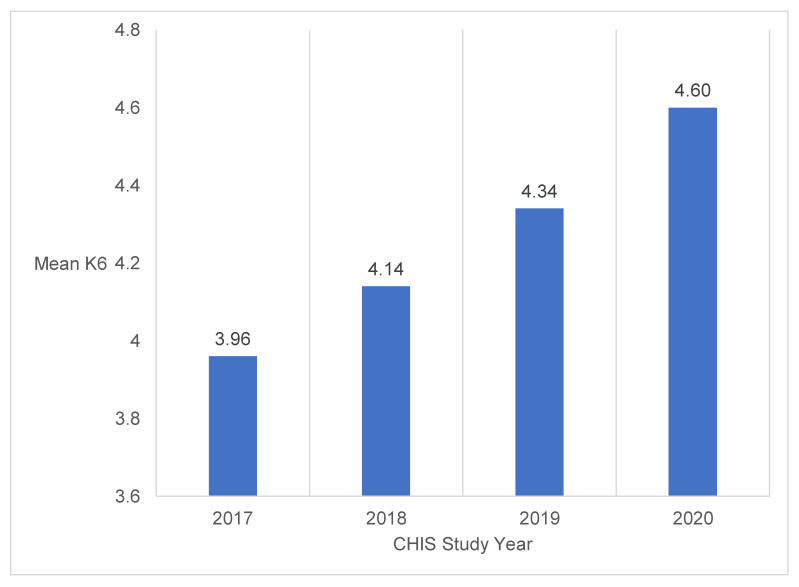
Mean Levels of Psychological Distress in CHIS 2017–2020.

**Figure 2 ijerph-20-00144-f002:**
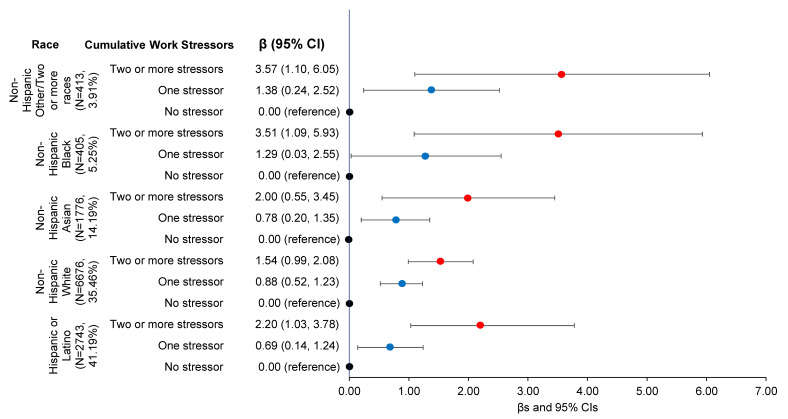
Associations of Cumulative Work Stressors with Psychological Distress by Race (βs and 95% CIs) (*N* = 12,113). Multivariable linear regression, adjusted for age, sex, marital status, educational attainment, household income, insurance, and citizenship.

**Table 1 ijerph-20-00144-t001:** Characteristics of the Sample Population in CHIS 2020, Weighted (*N* = 12,113).

Variables (*N*, %)	Hispanic or Latino (*N* = 2743, 41.19%)	Non-Hispanic White (*N* = 6676, 35.46%)	Non-Hispanic Asian (*N* = 1776, 14.19%)	Non-Hispanic Black (*N* = 405, 5.25%)	Non-Hispanic Other/Two or More Races (*N* = 413, 3.91%)	Total (*N* = 12,113)
Sex						
Male	1138 (53.98)	3178 (53.92)	871 (50.58)	162 (53.17)	180 (53.11)	5529 (53.40)
Female	1605 (46.02)	3598 (46.08)	905 (49.42)	243 (46.83)	233 (46.89)	6584 (46.60)
Age						
18–34	862 (41.30)	967 (29.65)	432 (35.32)	64 (26.47)	114 (41.68)	2448 (35.55)
35–49	1021 (34.58)	2122 (30.03)	641 (33.50)	112 (29.26)	145 (30.36)	4041 (32.37)
50–64	722 (21.00)	2534 (29.74)	556 (26.00)	173 (36.93)	112 (19.66)	4097 (25.59)
65+	138 (3.12)	1144 (10.58)	147 (5.18)	56 (7.34)	42 (8.30)	1527 (6.49)
Marital status						
Married	1362 (47.29)	4008 (54.67)	1109 (59.43)	157 (39.12)	201 (42.87)	6837 (51.02)
Widowed/Divorced/ Living with partner	627 (22.51)	1737 (24.41)	264 (13.34)	126 (25.47)	110 (24.71)	2864 (22.13)
Never married	754 (30.20)	1031 (20.92)	403 (27.24)	122 (35.41)	102 (32.42)	2412 (26.85)
Educational attainment						
University degree or higher	1247 (29.85)	4641 (60.27)	1409 (71.30)	230 (45.79)	256 (52.42)	7783 (48.24)
Some college	892 (22.73)	1624 (19.86)	256 (12.11)	129 (27.07)	124 (25.31)	3025 (20.53)
High school or less	604 (47.42)	511 (19.87)	111 (16.59)	46 (27.14)	33 (22.27)	1305 (31.23)
Household income (annual U.S. dollars)						
<50,000	909 (41.48)	1002 (16.77)	325 (22.89)	163 (35.89)	84 (17.80)	6818 (43.23)
50,000–99,999	818 (29.59)	1634 (25.26)	400 (23.50)	125 (29.70)	120 (40.41)	3330 (28.38)
≥100,000	1016 (28.93)	4140 (57.97)	1051 (53.61)	117 (34.41)	209 (41.79)	2552 (28.39)
Insurance						
Insured	2511 (88.15)	6606 (96.39)	1696 (94.53)	387 (90.90)	399 (94.07)	11599 (92.35)
Uninsured	232 (11.85)	170 (3.61)	80 (5.47)	18 (9.10)	14 (5.93)	514 (7.65)
Citizenship						
U.S. born citizen	1793 (55.80)	6225 (90.05)	592 (29.56)	362 (85.72)	360 (83.03)	9332 (66.86)
Naturalized citizen	625 (22.47)	406 (6.80)	885 (45.00)	31 (9.76)	47 (13.45)	1994 (19.09)
Non-citizen	325 (21.73)	145 (3.15)	299 (25.44)	12 (4.52)	6 (3.52)	787 (14.05)
Work stressors						
Job loss	270 (11.69)	439 (7.38)	134 (10.13)	36 (10.48)	32 (11.74)	911 (9.88)
Reduced work hours	623 (24.54)	1494 (22.22)	327 (20.65)	91 (22.50)	94 (20.26)	2629 (22.89)
Working from home	855 (22.46)	2647 (38.62)	747 (39.53)	151 (34.00)	166 (33.77)	4566 (31.66)
Cumulative work stressors						
None	1203 (48.62)	2753 (39.80)	697 (38.35)	160 (40.44)	163 (43.08)	4976 (43.39)
One	1346 (44.49)	3495 (52.66)	961 (53.70)	216 (53.06)	210 (48.34)	6628 (49.29)
Two or more	194 (6.89)	528 (7.54)	118 (7.95)	29 (6.50)	40 (8.58)	909 (7.32)
Psychological distress (mean, SE)	4.83 (0.14)	4.53 (0.09)	4.38 (0.15)	3.63 (0.30)	4.92 (0.31)	4.60 (0.07)

**Table 2 ijerph-20-00144-t002:** Associations of Work Stressors with Psychological Distress (βs and 95% CIs) (*N* = 12,113).

	Number of Exposed Participants	Model I	Model II
Job loss			
No	11,202	0.00	0.00
Yes	911	1.30 (0.70, 1.90)	1.18 (0.60, 1.76)
Reduced work hours			
No	9484	0.00	0.00
Yes	2692	1.00 (0.65, 1.34)	0.87 (0.53, 1.22)
Work from home			
No	7546	0.00	0.00
Yes	4566	0.72 (0.45, 1.00)	0.81 (0.53, 1.09)
Cumulative work stressors			
No stressors	4976	0.00	0.00
One stressor	6228	0.81 (0.53, 1.09)	0.80 (0.51, 1.09)
Two or more stressors	909	2.08 (1.48, 2.67)	1.98 (1.41, 2.56)

CI, confidence interval. Multivariable linear regression. Model I: adjustment for age and sex. Model II: Model I + additional adjustment for marital status, educational attainment, household income, insurance, and citizenship.

**Table 3 ijerph-20-00144-t003:** Associations of Work Stressors with Psychological Distress by Race (βs and 95% CIs) (*N* = 12,113).

Race	Work Stressors	Number of Exposed Participants	Model I	Model II
Hispanic or Latino (*N* = 2743, 41.19%)	Job loss			
	No	2473	0.00	0.00
	Yes	270	1.24 (0.20, 2.28)	1.17 (0.16, 2.18)
	Reduced work hours			
	No	2120	0.00	0.00
	Yes	623	1.23 (0.58, 1.88)	1.08 (0.43, 1.73)
	Work from home			
	No	1888	0.00	0.00
	Yes	855	0.42 (−0.15, 0.99)	0.44 (−0.18, 1.06)
Non-Hispanic White (*N* = 6676, 35.46%)	Job loss			
	No	6337	0.00	0.00
	Yes	439	1.35 (0.55, 2.15)	1.24 (0.49, 1.99)
	Reduced work hours			
	No	5282	0.00	0.00
	Yes	1494	0.78 (0.36, 1.21)	0.65 (0.24, 1.06)
	Work from home			
	No	4129	0.00	0.00
	Yes	2647	0.65 (0.32, 0.97)	0.79 (0.47, 1.11)
Non-Hispanic Asian (*N* = 1776, 14.19%)	Job loss			
	No	1642	0.00	0.00
	Yes	134	0.99 (−0.24, 2.23)	0.89 (−0.33, 2.11)
	Reduced work hours			
	No	1449	0.00	0.00
	Yes	327	1.05 (0.25, 1.86)	1.04 (0.24, 1.83)
	Work from home			
	No	1029	0.00	0.00
	Yes	747	0.96 (0.33, 1.58)	0.93 (0.31, 1.56)
Non-Hispanic Black (*N* = 405, 5.25%)	Job loss			
	No	369	0.00	0.00
	Yes	36	1.85 (−0.49, 4.19)	1.95 (−0.35, 4.25)
	Reduced work hours			
	No	314	0.00	0.00
	Yes	91	0.33 (−0.82, 1.49)	0.38 (−0.81, 1.57)
	Work from home			
	No	254	0.00	0.00
	Yes	151	2.21 (1.02, 3.40)	2.62 (1.14, 4.09)
Non-Hispanic Other/Two or more races (*N* = 413, 3.91%)	Job loss			
	No	381	0.00	0.00
	Yes	32	2.60 (0.44, 4.77)	2.69 (0.51, 4.88)
	Reduced work hours			
	No	319	0.00	0.00
	Yes	94	1.28 (−0.05, 2.61)	1.08 (−0.23, 2.38)
	Work from home			
	No	247	0.00	0.00
	Yes	166	1.47 (0.17, 2.76)	1.51 (0.23, 2.78)

CI, confidence interval. Multivariable linear regression. Model I: adjustment for age and sex. Model II: Model I + additional adjustment for marital status, educational attainment, household income, insurance, and citizenship.

## Data Availability

The raw data presented in this study are openly available at the UCLA Center for Health Policy Research at https://healthpolicy.ucla.edu/chis/data/public-use-data-file/Pages/public-use-data-files.aspx (accessed on 29 November 2022). The statistical SAS syntax supporting the conclusions of this article will be made available by the authors, without undue reservation. Requests to access the statistical SAS syntax should be directed to Jian Li, jianli2019@ucla.edu.

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
