# Peer review of "Associations of COVID-19 Related Work Stressors with Psychological Distress: Racial and Ethnic Disparities in Californian Workers"

_ijerph, 2022, doi:10.3390/ijerph20010144_

Round 1
Reviewer 1 Report
Thanks for inviting me to review this manuscript. This is a very interesting, well-organised, and insightful paper. I have only one major comment:
I would like to see an additional discussion about other work stressors and ethnic disparities in work-related work stressors and their (potential) psychological impacts in other contexts. This study considered three main work stressors: job loss, reduced work hours, and working from home. There are other important work stressors that might influence workers’ mental wellbeing, for example, stressor frameworks such as the transactional stress model (Lazarus & Folkman, 1984) and the job demand-control-support model (Karasek & Theorell, 1990). In addition, there are some more special stressors during Covid, such as the perceived risk of infection (e.g., Kirkman, 2021). How may these stressors influence workers of different ethnicities differently? Would they influence workers' mental wellbeing in the same way?
Also, there might be some more context-specific stressors. For example, in the Chinese context, ethnically discriminated-against migrants may suffer from higher job insecurity (Liu et al., 2022). They may not necessarily lose their jobs but were discriminated against during work. So they fear reporting their illness and have to conceal some particular symptoms. What would your research findings tell researchers in other contexts?
Reference
Karasek, R. T., & Theorell, T. T. (1990). Healthy work: Stress, productivity and the reconstruction of working life. New York, NY: Perseus Books Group.
Kirkman, J. (2021). Depressive Symptoms, Perceived Risk of Infection, and Emotional Fatigue among COVID-19 Frontline Medical Personnel. Psychosociological Issues in Human Resource Management, 9(1), 47-57.
Lazarus, R. S., & Folkman, S. (1984). Stress, appraisal, and coping. New York, NY: Springer Publishing Company.
Liu, Q., Liu, Z., Kang, T., Zhu, L., & Zhao, P. (2022). Transport inequities through the lens of environmental racism: rural-urban migrants under Covid-19. Transport policy, 122, 26-38.
Reviewer 2 Report
• Authors in the Introduction should briefly address the consequence of COVID-19 on the disruption to normal processes beyond just those pertained to quotidian life. Particularly, for authors it is necessary to capture in part the impacts of this health emergency across multiple socio-economic facets as to scope potential determinants and contributing factors that may influence work stressors (doi.org/ 10.3390/su14159699).
• Authors provided evidence of an association between work stressors especially among those identified as Black and those that had one or two/more stressors, with psychological distress. However, looking at the composition of the sample explored, these two categories appear limited in number (Blacks: 5.25%, individuals with two or more work stressors: 7.32%). Authors should discuss this inherent limitation of their study and outline how statistical significance observed here could be amplified due to the presence of large invariances in terms of composition.
• Authors note that the mean level of psychological distress was 4.60 in their sample, a notably higher score compared to previous years recorded. By contrast, those identified as Blacks appear to have a mean score of 3.63 which appears to be lower than any score observed overall from 2017. Authors should discuss these implications and what they could pose on the meaningfulness of their results, considering that their sample appears proportionally skewed against those with minimal (or lower) psychological distress. Authors here have an opportunity to address whether any pertained covariates were controlled for in their analysis.
Round 2
Reviewer 1 Report
I am satisfied with the current manuscript. Thanks for sharing.
Reviewer 2 Report
The authors have satisfactorily addressed most of my concerns.